# Aspects of Prevention of Urinary Tract Infections Associated with Urinary Bladder Catheterisation and Their Implementation in Nursing Practice

**DOI:** 10.3390/healthcare10010152

**Published:** 2022-01-13

**Authors:** Jitka Krocová, Radka Prokešová

**Affiliations:** 1Faculty of Health and Social Sciences, University of South Bohemia in České Budějovice, 370 11 Czech Budejovice, Czech Republic; rprokes@zsf.jcu.cz; 2Department of Nursing and Midwifery, Faculty of Health Care Studies, University of West Bohemia, 301 00 Pilsen, Czech Republic

**Keywords:** prevention, nursing, care quality, urinary infections, urinary bladder catheterisation, documentation

## Abstract

In the case of the prevention of catheter-associated urinary tract infections (CAUTI) related to healthcare provision, high-quality and comprehensively provided nursing care is essential. Implementation of preventive strategies is based on recommended procedures, and the introduction of whole sets of measures has been shown to be effective. The objective of this research is to find out whether the providers of acute bed care have implemented the steps of CAUTI prevention, and specifically which measures leading to improved quality of care in the area of urinary infections are already in place. To determine this, we carried out quantitative research. Data were collected using a questionnaire-based investigation; we used two non-standardised and one standardised questionnaire, and the respondents were general nurses in management positions (*n* = 186). The results revealed that result-related CAUTI indicators are monitored by only one-third of the respondents, and records of catheterisation indication are not kept by 17.3% of general nurses. The results of the research showed deficiencies in the monitoring of CAUTI outcome and process indicators, and a weakness of the implemented preventive measures is the maintenance of catheterisation documentation. Periodic CAUTI prevention training is not implemented as recommended. It is positive that there are well-working teams of HAI prevention experts in hospitals.

## 1. Introduction

Urinary tract infections associated with healthcare are the most common infection related to the provision of healthcare (HAI: healthcare-associated infections) [1,2,3,4,5]. The success rate in prevention of these infections, however, is relatively high [6,7]; therefore, of essential importance is the provision of high-quality and safe nursing care. Effective prevention steps include risk management in the area of CAUTI prevention, the introduction of nursing procedures/standards of care and their checks, the education of competent healthcare workers, respecting the relevant indication for catheterisation, the keeping of documentation on catheterisation, and cooperating with teams of experts in the prevention of HAI [8]. An overview study by Krocová and Prokešová (2019) summarises the conclusions reached in the research of the effectiveness of measures in the area of urinary tract infection prevention associated with healthcare published over the period between 2011 and 2018 [9] The review also includes a specification of nursing care in this area. The research presented in this article was performed based on the overview study [9]. This article presents research whose objective was to determine whether hospitals providing acute inpatient care in the Czech Republic have implemented CAUTI prevention measures. The results of the research published in this article are concerned with specific areas of prevention of infections associated with urinary bladder catheterisation. We present particular outputs which map the performance of checks and the implementation of steps aimed at improving quality of care in the area of urinary infections. We also wanted to determine whether risks of urinary infections related to healthcare are managed, and to verify whether activities targeting improvement of quality and care safety related to CAUTI prevention are performed. At the time the research began, there was no uniform CAUTI prevention procedure in place in the Czech Republic. Of pivotal importance in the case of CAUTI prevention are the recommendations contained in the Guideline for Prevention of Catheter-Associated Urinary Tract Infections [10]. This document, published by the Healthcare Infection Control Practices Advisory Committee (HICPAC) and the Centers for Disease Control and Prevention (CDC) [11] in 2009 and subsequently updated in 2019, recognises the verified steps of CAUTI prevention based on evidence-based practice (EBP). So-called reminders of the re-evaluation of indications for catheterisation are included in electronic patient health records [1,3,12,13,14,15,16]. Another topic to consider is the introduction of schemata and suggestions regarding the relevant indications for catheterisation, and systems of reminders to re-evaluate indication for catheterisation. Bernard et al. (2012) state that it is expected that nursing personnel have correct knowledge of relevant indications, and further mentions that programmes for re-evaluation of catheterisation should be introduced as a matter of course [17]. In their studies, Underwood (2015) and Yatim et al. (2016) describe the implementation of the HOUDINY protocol (haematuria, obstruction, urological surgery/intervention, decubitus ulcer, input/output monitoring, nursing care, and immobility), which contains indications for catheterisation and principles of CAUTI [5,18] prevention. The CDC recommendations (2009) concerned with the correct technique of urinary catheter introduction again mention the process of considering the relevancy of indication of catheterisation, utilisation of the technique of intermittent catheterisation, use of the smallest catheter size, aseptic catheterisation procedures that use one-way sterile packs, and the safety of the intervention when carried out by competent healthcare workers [10]. As part of the study, the results of which were published by Alexantis and Broome (2014), an education programme was implemented that included rehearsal of the skills, with concurrent implementation of a protocol of catheter care [19]. Subsequent audits confirmed a 98% agreement of meeting the standards and the procedures of nursing care, and an overall improvement in comprehensive care from 85.9% to 90.1%. In the area of correct urinary catheter care, the Guideline for Prevention of Catheter-Associated Urinary Tract Infections (2009) places emphasis on correct manipulation with the urinary collection system and hygienic care [10]. In relation to care and manipulation with the urinary collection system, an interesting study was published by Powers (2016), whose objective was to establish whether there is a relationship between the occurrence of CAUTI and the manipulation and disconnection of the urinary collection system, in the event that the aseptic procedure of disconnecting the system is performed according to standards [20]. In the setting of long-term catheterisation, the CDC (2009) recommends a silicone catheter, and correct hand hygiene is also an important part of the care [10]. The use of smaller catheter sizes (CH 12–14) is described [1] as a component of the CAUTI preventive measures in a study published by Andreessen et.al. (2012). Programmes and strategies for quality improvement recommend the introduction of standard procedures of care and their control as another effective preventive measure. The introduction or updating of standards of care as part of preventive procedures is described in many studies [2,18,20,21,22], as is the performance of audits of compliance with catheterisation procedures, care for patients with catheters, and CAUTI prevention.

The recommendation in the Guideline for Prevention of Catheter-Associated Urinary Tract Infections (2009) concerning documentation includes the introduction of healthcare records, ideally in electronic form, and the introduction of unified requirements for the records on catheterisation. The document also states that it is appropriate to use the clinical system to check the re-evaluation of indications for catheterisation and continuing education of non-physicians [10]. Education of competent healthcare workers is another recommendation of prevention; education should be performed at entry and periodically thereafter. The effect of comprehensive educational programmes is evaluated in the study by Viner (2020) [23] and Freeman-Jobson et al. (2016) [24].

The CDC (2009) evaluates the introduction of a standardised method of CAUTI monitoring as a strong recommendation supported by its moderate and high benefit in clinical practice, for monitoring CAUTI where tracking the number of CAUTI/1000 catheter days is recommended, as well as the number of secondary blood infections originating in the urinary tract and the number of catheter days [10]. Prevention of CAUTI does not only involve effective interventions, but also evaluations and monitoring of the number of infections [25]. In their review, Burston et al. (2013) present an overview of quality indicators monitored as part of nursing care which includes the processing of 40 studies and designates 43 verified quality indicators of nursing care [26]. The number of urinary tract infections as a quality indicator is reported in thirteen studies. Burston et al. (2013) mention that it is necessary to select, monitor, and evaluate sensitive indicators of the quality of care in connection with the monitoring of quality indicators [26].

## 2. Materials and Methods

### 2.1. Material

The research focused on a specific area of nursing care; respondents were non-physician healthcare workers occupying the positions of upper and middle management in hospitals providing acute bed care in the Czech Republic. We contacted non-physician healthcare management in 194 hospitals providing acute bed care in the Czech Republic and asked them to give their consent to the performance of the research. After three rounds of communication, agreement with the research was granted by the management of 34 hospitals, and all three of the questionnaires were completed by 186 respondents. Based on this consent, we distributed questionnaires to leading non-physicians (general nurses) in the positions of head, ward, and leading nurse. The selection of respondents corresponded to the research intension, i.e., assessment of the implementation of CAUTI preventive measures in nursing practice. On the basis of the competencies given by the legislation, general nurses were targeted. General nurses are competent to care for the urinary catheters of patients of all ages, including bladder lavages and bladder catheterisation in women and girls older than 3 years. After obtaining special professional competence (certified courses) or specialised “intensive care” competence, they are also competent to perform catheterisation in men.

### 2.2. Methods

The study was performed using quantitative research, and the subsequent data analysis was conducted using methods of mathematical statistics. The research instruments were three questionnaires which were modified to enable online data collection. As part of the research, a standardised questionnaire was used that is currently employed in the DUQuE (Deepening our understanding of quality improvement in Europe) project, questionnaire D: Systems of quality improvement in European hospitals (questionnaire for quality managers/coordinators). The DUQUE project was financially supported by the 7th Framework Programme of the European Union (EP7/2007–2013) as part of grant agreement No. 241822. The use of the questionnaires was approved by the project coordinator and head of research of the DUQuE project, Prof. Rosy Sunol. The questionnaires are available in the Czech language and were modified for use in our research, i.e., our own questions were added which accepted the intentions of the research, and the questionnaires were shortened. The questionnaires modified in this way were already used in the Czech Republic by PhDr. Radka Pokojová, Ph.D. (2018) who agreed to their use for data collection [27]. The modification or completion of the questionnaires consisted of adding questions concerned with the identification of characteristics related to the performance of the respondents’ profession: work position, type of workplace, and hospital.

Non-standardised questions were designed to establish whether the selected methods of quality improvement in nursing care were implemented in the hospitals in the Czech Republic with a focus on CAUTI prevention. When designing the questionnaires, or the questions in the non-standardised questionnaires, individual areas (dimensions) of prevention that are determined to be effective based on EBP were specifically targeted. The use of standardised questionnaires focusing on quality and safety management in general, and non-standardised questionnaires focusing on a specific area of CAUTI prevention, was intentional; the aim of the research was to perform a comprehensive assessment of the situation in the broader context of the general principles of quality management and control.

At the beginning of the research a pilot phase was performed in four hospitals, and based on the obtained recommendations, the questions in the non-standardised questionnaires were modified. Due to the online distribution of the questionnaires, a trial of data export and its relevancy was verified by a statistician. The research was conducted between March 2020 and December 2020.

Data analysis was conducted with the programmes SASD 1.5.8 (Statistical data analysis) and SPSS, based on first- and second-degree classifications. In the first-degree classification, frequency tables were constructed for individual measures and then absolute and relative frequencies and mean values were calculated. In the second-degree classification, contingency tables were constructed containing absolute and relative frequencies and the sign schema. As part of the relationship analysis, a chi-square test of goodness of fit—x^2^ (Pearson chi-Square) and an independence test were applied, according to the characteristic of the signs and the number of observations. Further steps included the calculation of the Pearson coefficient of contingency, normed Pearson coefficient of contingency, Čuprov coefficient, Cramer’s coefficient, Wallis coefficient, Spearman coefficient, and correlation coefficient. The strength of the relationships was measured on three levels of significance: α = 0.05, 0.01, and 0.001. As part of the description of the analysed statistically significant bonds, the values of the chi-square test of goodness of fit and the independence test were given as standard. The level of possible deviation was calculated for each cell of the contingency table, which enabled us to determine the direction of the statistically significant relationship between two characteristics. The study did not require ethical approval from an institutional review board. Due to the topic of our research, it was not necessary to seek ethical approval. The management of the hospitals where the respondents work gave their approval to collect the data.

## 3. Results

It is the purpose of this article to present the results of the research which mapped the implementation and performance of CAUTI preventive measures. The prevention of urinary tract infections involves multifactorial measures, or the introduction of more measures. For the sake of clarity, the presented results have been divided into eight areas. Table 1 shows the frequencies of responses by variable hospital type. According to the type of ward, there were 111 (*n* = 111) respondents from the internist ward type, 65 (*n* = 65) from the surgical ward type, and 10 respondents were in the position of nursing care assistant.

### 3.1. Indication for Catheterisation

More than half of the respondents fully agreed with the statement that the indication for urinary bladder catheterisation is always recorded in patient documentation, while 30% of the respondents partially agreed and 17.3% disagreed (Wallis coefficient = 1.37206; Spearman’s coefficient = −0.27692; correlation coefficient = −0.261618). Nurses confirmed clearly determined indications (e.g., by a standard, internal regulation) for urinary bladder catheterisation in 58% of cases, while partial agreement was given by 17.3% (x^2^= 7.62413; *p* = 0.0221025). Agreement with the statement “I can specify the indications for catheterisation according to current recommended procedures” was given by 64.7% of nurses, partial agreement by approximately one-quarter, and disagreement was expressed by 10% of the respondents. A total of 60% of the nurses believe that as general nurses they can voice their opinion on the indication of urinary bladder catheterisation in their patients. A total of 50.8% of nurses agreed and 40.7% partially agreed that if they express their opinion about the urinary catheterisation, their view is respected by the physician. The statement “Indication for the catheterisation is a facilitation of nursing care” resulted in the agreement or full agreement of more than one-third of the respondents and the partial agreement of 24.1% of nurses. The results are demonstrated in Figure 1.

In this connection it also of interest that general nurses who believe that they can voice their opinion or are competent to evaluate the indication for urinary bladder catheterisation state significantly more often that if they express their opinion it is respected by the physician (x^2^ = 84.445; *p* ≤ 0.001). Table 2 provides the responses to the questionnaire items related to the area of “competency to evaluate indications for catheterisation and the physician respects their opinion”.

We further demonstrated a statistically significant relationship between the type of workplace and whether a standard of indication is determined regarding urinary bladder catheterisation. It holds true that this standard is significantly more frequently determined at non-surgical workplaces (x^2^ = 7.624; *p* < 0.05). Furthermore, it was demonstrated that records of the indication for bladder catheterisation are kept markedly more often in non-surgical workplaces (x^2^ = 6.099; *p* < 0.05).

### 3.2. Education

In this area, a statistically significant relationship was shown between the support of education by executive workers and the possibility to take part in education during working hours (x^2^ = 105.237; *p* < 0.001). It unequivocally holds true that where education is supported by the upper management, healthcare workers can participate in education during working hours. It was interesting to compare the approach to non-physician education by ward type.

In the area of correct procedures of urinary bladder catheterisation, care for patients with urinary catheters, and the possibilities of prevention of infections that are related to healthcare, non-surgical workplaces provide education as part of entry training to 20.0% of nurses, compared to 23.1% of nurses in surgical workplaces. Periodic education activities are performed on a regular basis by approximately 38.5% of nurses, both in surgical and non-surgical workplaces. A difference was recorded in the case of the answer “other”, which related to education in the form of a certified course or specialised education; this option was stated by 28.3% of nurses from non-surgical workplaces and 17.3% of nurses from surgical workplaces.

### 3.3. Increasing the Quality of Care and Care Quality Monitoring

In this area, a statistically significant relationship was demonstrated between hospital type and the appointment of one or more quality and safety managers/coordinators (x^2^ = 15.013; *p* < 0.05; Wallis coefficient = −15.6022). It holds true that these managers/coordinators are appointed to a significantly lesser extent in municipal-type hospitals. The strength of the test was limited by an insufficient number of observations in six cells of the contingency table. We applied Yate’s correction. A statistically significant connection between hospital type and the existence of acknowledgements/incentives to increase the quality of care was demonstrated (x^2^ = 21.295; *p* < 0.001). It holds true that these acknowledgements/incentives are implemented significantly more in private-type hospitals and other hospitals, and significantly less in municipal-type hospitals. The strength of the test was reduced by an insufficient number of observations in one cell of the contingency table. Yate’s correction was applied. In other cases where the analysis was performed based on second-degree classification, the tests did not demonstrate a statistically significant relationship between hospital type and the characteristics given in Table 3 and Table 4. Table 3 and Table 4 provide the responses to the questionnaire items related to the area of “improving and increasing the quality of care and care quality monitoring”.

The statement that healthcare workers are trained in procedures ensuring patient safety was agreed with or mostly agreed with by 88.4% of the respondents, and the claim that healthcare workers are supported to report accidents and undesirable events was agreed with by 90% of the respondents.

The research shows that where regular audits of the procedure of urinary bladder catheterisation are performed, workplaces are more likely to have implemented a standard for the procedure of urinary bladder catheterisation (x^2^ = 32.970; *p* < 0.001). Table 5 provides the responses to the questionnaire items related to the area of “existence of a standard of care and regular audits”.

The same finding was made in the case of the evaluation of the existence of a standard for the prevention of urinary tract infections and the performance of regular audits of acceptance of preventive CAUTI measures. Furthermore, it was determined that where regular audits of the prevention of urinary tract infections are performed, a standard of prevention of urinary tract infections is significantly more likely to be implemented (x^2^ = 23.171; *p* < 0.001).

### 3.4. Risk Management

The question “Does your facility evaluate the risk of urinary tract infections related to an introduced urinary catheter?” was answered positively by 26.8% of the respondents and negatively by 73.2%. Other questions targeted the monitoring of result-related quality indicators; the results showed that the number of cases of urinary infection related to catheterisation is not monitored, according to the answers of 66.7% of the respondents. The number of cases of secondary blood infections that have their source in the urinary tract is not monitored by 64.5%, and the number of catheter days (related to the number of nursing days and expressed in %) is not monitored, according to three-quarters of the nurses asked.

### 3.5. Support of IT Technologies in the Process of Quality Improvement and Care Safety and Risk Management

It was found in this area that the approach of individual hospitals by type is homogeneous and does not statistically significantly differ. No statistically significant relationship was demonstrated between hospital type and the characteristics related to the given area of CAUTI prevention. Table 6 provides the responses (relative frequencies) to questions concerning the introduction of IT technologies as measures for increasing the quality and safety of care and risk management.

According to the respondents, a supportive system, such as system of reminders and notes, is implemented in 10.5% of cases. The keeping of electronic patient documentation was confirmed by 49.2% of nurses. The existence of the function “reminder/reminders for re-evaluation of the indication of urinary bladder catheterisation” as part of electronic healthcare documentation was agreed with by 10.8% of nurses.

In this dimension, it was further demonstrated that workplaces that have reminders of the catheterisation indication as part of their electronic documentation perform regular re-evaluations of the indication for catheterisation once in 24 h significantly more often (Table 7).

### 3.6. Documentation of Urinary Bladder Atheterisation

In this area, the keeping of documentation on urinary bladder catheterisation was evaluated. The statement “The type and size of the urinary catheter is always recorded in the patient documentation” was agreed with or fully agreed with by 88.7% of the respondents. The keeping of records on performed hygienic care was confirmed by 60.7%, partially confirmed by 16%, and recording of this data was not confirmed by 23.3% of the respondents (Spearman’s coefficient = −0.0928426, correlation coefficient = −0.0924602). The correctness of the keeping of medical records on urinary bladder catheterisation was audited according to 87.3% of the respondents. In the area of record keeping, a relationship was demonstrated between the type of workplace and regular records of the catheterisation data in patient documentation. It holds true that the date of catheterisation is recorded in the documentation in a significantly higher proportion of cases (x^2^ = 6.505; *p* < 0.05) at internal-medicine-type wards. An identical finding also holds true for the recording of complications of catheterisation (x^2^ = 6.505; *p* < 0.05) and the entry/recording of the indication for catheterisation (x^2^ = 12.545; *p* < 0.01).

### 3.7. Consumables and Aids

We determined a statistically significant relationship between workplace type and the possibility to select from the recommended catheter sizes (x^2^ = 6.671; *p* < 0.05). It holds true here that this possibility is significantly greater at internal medicine wards. Another statistically significant relationship was between non-surgical workplaces and the possibility to select the types of urinary collection systems. A significantly greater choice of the types of urinary collection systems was again demonstrated in internist wards (x^2^ = 6.952; *p* < 0.05). It is nonetheless necessary to add to the above that the sufficient availability of consumables for the procedure of urinary bladder catheterisation was confirmed by 96.7% of the respondents. The statement that the workplace offers a selection of catheters in terms of material (latex/silicon, silicon, other) was agreed with or fully agreed with by 68.7% and partially agreed with by 16.0% of the respondents.

### 3.8. Teams of Experts

In this area, the dependence between hospital type and indicators targeting the functioning and activity of a team of experts for the prevention of infections related to healthcare in hospitals was examined.

The approach of individual types of hospital is homogeneous and does not statistically significantly differ in these areas. Table 8 shows the responses to the questionnaire concerning the existence and functioning of a team of HAI prevention experts. Correctly functioning teams for HAI prevention in the hospital were confirmed by 74.1% of the respondents, and approximately one-half of the respondents agreed or fully agreed with the statement that there is an employee entrusted with checking the measures of HAI prevention (Wallis coefficient = −1.40684). In this area, it was determined that workplaces where a team of experts for infection prevention exists are more likely to appoint an employee entrusted with infection prevention (x^2^ = 45.230; *p* < 0.001). The research also showed that workplaces where a team of experts for infection prevention exists conduct regular audits headed by the respective authorised worker significantly more often (x^2^ = 78.660; *p* ≤ 0.001).

The presence of an expert for the prevention of HAI who conducts regular audits of care, and the observance of hygienic-epidemiological procedures at the workplace was confirmed (agreed or fully agreed) by 66.3% of the respondents and partially confirmed by almost 14.0%. Electronical reporting of HAI was not confirmed by 26.5% of the respondents.

## 4. Discussion

The results presented in this article map the introduction of CAUTI prevention measures by acute bed care providers in the Czech Republic. In the discussion, the results, according to the dimensions of CAUTI prevention, will be compared with the results of other studies. The Guideline for Prevention of Catheter-Associated Urinary Tract Infections (2009) clearly sets out the relevant indications for urinary bladder catheterisation, and at the same time presents cases of improper catheter introduction [10]. The results of the presented research show that indications for catheterisation determined in writing were confirmed by 60% of the respondents. In these cases, the indications are usually determined in a standard procedure concerning the area of urinary bladder catheterisation. In this context, the authors of the article note that the management of the exact indication for bladder catheterisation as part of the documentation is reported in many studies; this measure seems to be very effective [5,14,18,19,28,29,30,31]. Oman et al. (2012) describe in their study [29] how a comprehensive programme of education for nurses was introduced—including information about indications—concurrently with the introduction of precise indications for catheterisation and the function of a reminder to re-evaluate the urinary bladder catheterisation as part of the electronic medical records [29]. The research further showed that the indication for catheterisation is part of the record according to one-half of the respondents, and roughly one-third agreed partially. Of interest was the finding that the indication is recorded significantly more frequently in non-surgical workplaces. Another finding made in the research was that half of the nurses confirmed, and 40% of nurses partially confirmed, that if they express their view about the indication then their opinion is respected by the physicians. Johnson et al. (2016) describe the introduction of a protocol with clear-cut criteria for the indication of catheterisation, with the possibility for nurses to remove the catheter; furthermore, the protocols determined indications when a nurse indicates the necessity of a catheter to remain introduced in some cases, e.g., disease or trauma of the urotract, chronic retention of urine, difficult catheterisation, gynaecological/urological surgery, or haematuria, the removal is decided by the physician [31]. An exception is a study by Kim et al., (2017) in which the only preventive intervention was a new system of documentation [32]. Catheterisation or indication for catheterisation for the “facilitation of nursing care” found little or no agreement in the case of 41% of nurses. In their review, Bernard et al. (2012) describe the topic of urinary bladder catheterisation without indication where the catheter is frequently introduced in the urgent admission ward and the indication is not further re-evaluated [17]. Education of competent healthcare workers is another measure in CAUTI prevention. Its effectiveness is shown especially in periodic courses [10] that are comprehensive and contain information about correct procedures of urinary bladder catheterisation, the issues and prevention of CAUTI, other complications of catheterisation, and alternatives of catheterisation. The CDC (2019) mentions this as a strong recommendation [10]; however, it is supported by weak evidence of a clear clinical benefit. It follows from the presented research that if the leading non-physician healthcare worker supports education, then he or she allows nurses to take part in these courses during working hours. The research further showed that periodic courses are attended by roughly 40% of nurses (no difference was detected between various types of workplaces). Jain et al. (2015) report that the education of healthcare workers in the area of CAUTI prevention holds high priority and is essential as a measure to minimise the occurrence of infections [33]. In his article, Viner (2020) highlights education in CAUTI prevention for employees in long-term care and states that trained nurses can educate other healthcare workers who participate in the care, e.g., physiotherapists, ergotherapists, or speech therapists working in long-term therapy [23].

The results concerning improvement and monitoring of the quality of care showed that the existence of a quality manager depends on the type of hospital. Overall, the position is established, according to 93.9% of the respondents. Incentives to improve the quality of care in the form of bonuses or acknowledgement were confirmed by 40.9% of the respondents. Feedback on the quality of patient care is received by 80% of the respondents. Carter et al. (2016) state that the motivation of hospital management to provide high-quality and safe care is unequivocally the prestige of the hospital, as well as the publishing of data that map the number of undesirable events, including infections and financial penalties from insurance companies [21]. McNeill (2017) reports that upper management should also provide space for members of nursing teams to voice their opinion about the results and their evaluation, and should prompt them towards proposing steps to improve the quality of care [7].

The research demonstrated a relationship between the regular performance of audits of CAUTI prevention procedures and the existence of a standard procedure of CAUTI prevention. Workplaces with an implemented standard perform the audits significantly more frequently.

It was shown that in the area of CAUTI risk management the result indicators, or the numbers of cases of CAUTI, are monitored according to one-third of the respondents, the number of catheter days is monitored according to 24.7% of the respondents, and the monitoring of secondary blood infections is performed according to 24.5% of the respondents. Monitoring of the above-mentioned result indicators is recommended by the Guideline for Prevention of Catheter-Associated Urinary Tract Infections (2009) [10]. The results from this area will be additionally verified as part of continuing research in the form of interviews with the quality managers of the providers of acute bed care.

It is recommended to use IT technologies in the prevention of CAUTI, or as part of the introduction and implementation of prevention programmes. The CDC (2019) recommends record keeping throughout the facility in the standard form, and exclusively electronically, especially due to the statistical processing of data related to CAUTI [10], ongoing audits of the records and monitoring of the indicators. This is also mentioned in the recommendation of the American Association of Critical-Care Nurses [6]. It follows from the research that the keeping of electronic records was confirmed by almost one-half of the nurses. The introduction of a reminder function for the re-evaluation of the indication for catheterisation was only confirmed by roughly 11% of the respondents. Relating to this, it was shown that the re-evaluation of catheterisation indication is more frequently performed in cases where the reminder function is included in the electronic documentation. Parry et al. (2013) report the results of a study where a hospital introduced unified documentation in electronic form, and the number of cases of CAUTI decreased by 70% after 36 months, while the number of catheter days dropped by 50% [34]. Markovic-Denic and Mijovič (2010) mention the necessity to use closed urinary collection systems, and to use antimicrobial-layer catheters in cases of chronic urinary tract infections [35].

The results of the research revealed that the statement “The workplace has a sufficient amount of consumables for catheterisation” was supported by 15.1% of the respondents, while full agreement was voiced by 81.9% of the respondents. The research further confirmed that internal-medicine-type workplaces report the possibility to select the sizes of urinary catheters and urinary collection systems significantly more often. More than one-half of the respondents (52.4%) fully agreed and 14.5% of nurses agreed with the statement that the workplace has a selection of catheters according to material type. Holroyd (2019) compares the advantages and disadvantages of materials, and calls attention to the possibility of allergies in the case of latex catheters [36]. The results of the presented research appear to be very positive in this area, with hospitals providing sufficient quantities of relevant consumables and catheterisation aids.

The research further determined whether the providers of acute bed care have functioning teams of experts for the prevention of CAUTI or HAI. Activity of multidisciplinary teams of HAI prevention is described in several contributions [1,4,28,32,34,37]. Members of these teams keep records of infections, check the quality of care, assist in the introduction of preventive measures, and play a fundamental role in the prevention of infections related to the provided care. The results of the research confirm the existence of teams for the prevention of HAI in hospitals according to 74.1% of the respondents, and almost one-half (49.4%) of the respondents agree or fully agree with the statement that there is an employee at the workplace entrusted with checking the HAI prevention measures. It was further proven that in the case of the existence of a team of experts at the workplace, a healthcare worker is more often appointed whose responsibility is the prevention of healthcare-associated infections. The research determined that HAI are reported in an electronic form, according to 60.9% of the respondents. The respondents further confirmed, or agreed or fully agreed, that the teams of experts perform audits of care and of HAI prevention in 66.0% of cases. Pintar (2013) reports [38] that based on EBP evidence, the introduction of the function of a worker in the area of infection prevention is a step towards increased safety and quality of care. The first step in the study published by Andreessen et al. (2012) was the establishment of a multidisciplinary team in the hospital [1]. Nevertheless, it follows from the above that the introduction of the steps of prevention and management of CAUTI prevention must always be comprehensive and accepted in cooperation with all competent healthcare workers, and must be checked on a continual basis. The research was performed with the aim of mapping the current situation regarding the prevention of CAUTI from the point of view of nursing care. The results of the research were provided to the management of the hospitals where the research was performed. The authors of the article were approached by the leading general nurses of three hospitals to help in the implementation of educational programmes related to the issue of CAUTI prevention. Based on the results of the study, the authors of the article recommend accepting and implementing the currently issued national nursing procedure for providers of acute inpatient care. The expected continuation of the study will focus on the mapping of the implementation of the national standard in hospital practises and the verification of its practical fulfilment.

## 5. Conclusions

The objective of the research was to find out whether the providers of acute bed care have implemented the steps of CAUTI prevention, and specifically, what measures leading to increased quality of care in the area of prevention of urinary infections are implemented and also checked. Nurses make up the most numerous group of healthcare workers, playing an essential role in care quality management and in the introduction of preventive measures and their checks [39]. Our research focused on the implementation of preventive measures recommended in the Guideline for Prevention of Catheter-Associated Urinary Tract Infections (2009) in clinical practice in the facilities of acute bed care providers. Acceptance of the relevant indications is a very strong preventive measure. In this area of prevention, the respondents confirmed that records are made of the indication into the patient medical records. This fact was not mentioned by only 17.3% of the respondents. Confirmation of the practice of recording data on urinary bladder catheterisations was given by more than 88% of the nurses conducting hygienic care; however, it was recorded by 60.7% of the respondents. Workplaces where the reminder of indication for catheterisation exists as part of the electronic documentation perform regular re-evaluations of indication significantly more often. Because of the above-mentioned facts, the authors recommend the following: “In this regard, the introduction of unified documentation as part of the clinical information system at the healthcare facility suggests itself as being of benefit, together with the function of reminders to re-evaluate the indication for catheterisation, etc.” Periodic education in the field of care for patients with urinary catheter and CAUTI prevention was confirmed by approximately 38% of nurses. It is important that the education programmes are comprehensive and that the education is supported by the management. Ensuring patient protection from HAI consists of the monitoring of the number of these infections. It is also essential to determine sensitive indicators of quality of care that must be monitored and evaluated. In our research, monitoring of the number of CAUTI cases was confirmed by 33.3% of the respondents, and monitoring of catheter days was confirmed by approximately 25% of the respondents. The introduction of standard procedures is another prevention step. The research showed that if the workplace uses an implemented standard of the procedure of urinary bladder catheterisation, then regular audits are performed significantly more often than in workplaces where there is no such standard of care. In contemporary nursing, the existence of standard procedures of care is regarded as being indispensable, as is the performance of audits of care. Standards are further regarded to be an instrument to improve the quality of care, and, related to that, it is the function of audits to provide feedback to quality managers about the efficacy of their management of nursing care [40]. The National Nursing Procedure “Urinary Bladder Catheterisation” [41] has been introduced in the Czech Republic, and managements of healthcare providers are obliged to modify their local standards and perform regular care audits in accordance with this document. The procedure was published at the time of our research, and its implementation in hospitals is ongoing. The research also showed the necessity for the existence of a team of experts, and it was confirmed that if teams of experts for infection prevention function in the facility, then there is also an employee entrusted with infection prevention. Kilíková and Jakušová (2008) state that the management of the activities of the whole system plays a directing role, from which the level of quality of care and the attained results of care are derived [40]. The implementation of CAUTI preventive measures must be comprehensive; however, in our case, the research has shown insufficient monitoring of quality outcome indicators.

Furthermore, there were shortcomings in catheterisation documentation where the recording of indications for catheterisation and records of hygienic care were neglected. Another weakness in CAUTI prevention is the failure to maintain documentation electronically as a part of the clinical information system. The authors of the article also point out the need to introduce periodic educational programmes. In addition, the research results show that there are enough consumables and catheterisation aids available in hospitals and there are teams of experts for HAI prevention in hospitals.

### Study Limitations

The authors see the limits of the research in the low level of willingness of the hospitals to participate in the research, even if performed in strict anonymity. Therefore, a limitation is the low number of respondents. This is directly related to the COVID-19 epidemiological situation.

## Figures and Tables

**Figure 1 healthcare-10-00152-f001:**
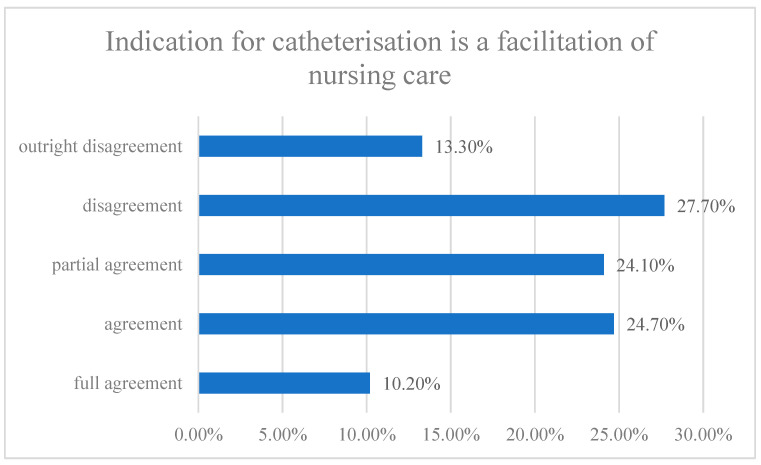
Overview of the responses to the statement “Indication for the catheterisation is a facilitation of nursing care”.

**Table 1 healthcare-10-00152-t001:** Frequency of responses according to the variable hospital type.

Hospital Type	Absolute Frequency	Relative Frequency
University	50	26.9%
Regional I	36	19.4%
Regional II	27	14.5%
Municipal	28	15.1%
Private	35	18.8%
Other	10	5.4%
Overall	186	100.0%

**Table 2 healthcare-10-00152-t002:** Responses to items related to “competency to evaluate indications for catheterisation and the physician respects their opinion”.

Competence to Assess Indications for Catheterisation and Acceptance of the Nurses Opinion-Assessment of Indications for Cathetrisation	Outright Disagreemen	Disagreement	Partial Agreement	Agreement	Full Agreement
Competency of A general nurse to voice HIS OR her opinion ON the indication of catheterisation and…	4.2%	3.6%	25.3%	37.3%	29.5%
the opinion of the general nurse regarding the indication of catheterisation IS RESPECTED by THE physician	5.4%	3.0%	41.0%	30.7%	19.9%

**Table 3 healthcare-10-00152-t003:** Responses to items related to “improving the quality of care and care quality monitoring”.

Improving Quality of Care and Monitoring Quality of Care	Yes		No	
	Absolute Frequency	Relative Frequency %	Absolute Frequency	Relative Frequency %
There is a special internal calculation for quality improvement	88	47.3	98	52.7
One or more control groups or committees are established	159	85.5	27	14.5
One or more quality/safety managers/coordinators are appointed	170	91.4	16	8.6
Acknowlegments/incentives for quality improvement exist	74	39.8	112	60.2

**Table 4 healthcare-10-00152-t004:** Responses to items related to “increasing the quality of care and care quality monitoring”.

Increasing the Quality of Care and Care Quality Monitoring	I Disagree	I Mostly Disagree	I Mostly Agree	I Agree
…Provision of feedback to patient care	2.2%	17.7%	30.9%	49.2%
…Is supported to report accidents and undesirable events	0.0%	9.9%	28.7%	61.4%
…Authorisations for activities are checked by the administrative body	1.7%	16.0%	26.5%	55.8%
Healthcare workers are trained in procedures ensuring patient safety	0.0%	11.6%	38.1%	50.3%

**Table 5 healthcare-10-00152-t005:** Responses to items related to “existence of a standard of care and regular audits”.

Existence of a Standard of Care and Regular Audits	Outright Disagreemen	Disagreement	Partial Agreement	Agreement	Full Agreement
Existence of standard of care for the procedure of urinary bladder catheterisation	1.8%	9.0%	2.4%	15.7%	71.7%
Regular audits of the procedure of urinary bladder catheterisation	14.5%	25.9%	23.5%	12.7%	23.5%

**Table 6 healthcare-10-00152-t006:** Introduction of IT technologies in the process of increasing the quality and safety of care and risk management.

Introduction of It Technologies in the Process of Increasing the Quality and Safety of Care and Risk Management	Introduced		Not Introduced	
	Absolute Frequency	Relative Frequency %	Absolute Frequency	Relative Frequency %
Reminder … in electronic form	18	10.8	168	89.2
Reminder … not in electronic form	58	36.8	128	63.2
Electronic patient medical records	89	49.2	97	50.8
Test results … in electronic form	62	34.3	124	65.7
Electronic drug prescription	79	43.6	107	56.4
Supportive systems (reminders, notes)	19	10.5	167	89.5

**Table 7 healthcare-10-00152-t007:** Relationship of the variables “re-evaluation of indication for catheterisation and the function of reminders for re-evaluation …”.

Regular Re-Evaluations of the Indication for Urinary Bladder Catheterisation Once in 24 h	*X*^2^ Value	*p*
“Reminder” of the evaluation of the indication of catheterisation as part of electronic documentation	21.096	<0.001
“Reminder” of the evaluation of the indication of catheterisation in other ways than electronically	37.507	<0.001

**Table 8 healthcare-10-00152-t008:** Responses to items related to the functioning of the HAI prevention expert or team and the form of HAI reporting.

Teams of Experts for the Prevention of HAI and Form of HAI Reporting	I Disagree	I Mostly Disagree	I Mostly Agree	I Agree	I Fully Agree
Functioning of team of experts for the prevention of hospital infections	4.9%	10.2%	10.8%	22.8%	51.3%
Authorised employee for prevention of infections at of the workplace	10.8%	27.7%	12.1%	17.5%	31.9%
Reporting of the occurrence of HAI in the form of a printed document	5.4%	24.7%	15.1%	24.7%	30.1%
Reporting of the occurrence of HAI in the form of an electronic form	16.8%	9.6%	12.7%	19.9%	41.0%
Employee of the ward/section entrusted with reporting the occurrence of HAI	8.4%	12.7%	8.4%	32.5%	38.0%
Occurrence of HAI is reported by the attending physician	3.6%	10.2%	18.1%	28.3%	39.8%
Regular audits are performed by an expert for HAI prevention	5.3%	14.5%	13.9%	22.3%	44.0%

## Data Availability

Research data are available on request from the authors.

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
