# Peer review of "Aspects of Prevention of Urinary Tract Infections Associated with Urinary Bladder Catheterisation and Their Implementation in Nursing Practice"

_healthcare, 2022, doi:10.3390/healthcare10010152_

Round 1

Reviewer 1 Report

Title: Aspects of Prevention of Urinary Tract Infections Associated with Urinary Bladder Catheterisation and their Implementation in Nursing Practice

This manuscript corresponds to the scope and purpose of the journal. The topic of the study chosen by the authors is important. It highlights the role of risk management in monitoring of quality of care indicators, and an education of competent healthcare workers. Authors also support an idea to employ IT technologies in areas of keeping records, monitoring of quality indicators, and production of educational programmes. I appreciate that this study brings new ideas in the field of nursing practice. However, data acquisition took time between March and December 2020 during a period of the COVID-19 pandemic. For this reason, the top management of several addressed Czech hospitals was less interested for implementation of such research. Hence, a lower number of respondents in article is only limiting factor of this study.

The authors should continue in their interesting research in order to increase the number of the respondents. Acceptable without revision.

Author Response

no comments

Reviewer 2 Report

Interesting work; A higher number of centers and patients are required. 

Main question addressed by the research is use of measure packages of CAUTI prevention in Czech Republic hospitals, using questionnaires answered by workers.

The conclusions are consistent with the evidence and arguments presented. Also, authors addressed the main question posed. References are appropriate.

The topic is relevant because CAUTI prevention application between health care centers is inconstant and irregular, so more study is needed.

This manuscript contributes to the subject area, since there is sparse data that has been published regarding this subject.

The study was performed using questionnaires oriented to the hospital workers. An objective analysis of the results of this prevention programs application is needed.

Author Response

Presentation of the results modified.

The analysis of the educational program (e-learning) is ongoing. The results will be supplemented by qualitative research (interviews with quality managers).

Reviewer 3 Report

Aspects of Prevention of Urinary Tract Infections Associated with Urinary Bladder Catheterisation and their Implementation in Nursing Practice

The word facilitation in graph 1 and text and also the question are difficult to understand; is this a ‘fact’ or an ‘opinion’ and how does this relate to local and national regulations? But also: What are ‘general nurses’? are these ‘basic level’ nurses? (that would not be allowed to indicate or perform urinary catheterisation without supervision by law in many healthcare systems.) And furthermore it is not at all clear who were the responders: [cited:] ‘we distributed questionnaires to leading non-physicians (general nurses) in the positions of head, ward, and leading nurse.’ But who responded?? Were all respondents ‘identical level’? (and discuss how did (or can) the professional level of the respondents have affected the results?

Also was the total number of respondents 186 or 176 (internist + surgical) and or why is this 186(or 176)/194 considered low level of willingness in the discussion? Or does this refer to a low number of responses (see below: number of responses not given.)

There is a lot of repetition in the text: I give one example but almost all paragraphs are identical: ‘The same finding was made in the case of the evaluation of the existence of a standard for the prevention of urinary tract infections and the performance of regular audits of acceptance of preventive CAUTI measures. A statistically significant relationship was demonstrated between whether the workplace has implemented a standard of urinary tract infection prevention and regular audits of urinary tract infection prevention (x2= 23.171 p<0.001). Where regular audits of the prevention of urinary tract infections are performed, a standard of prevention of urinary tract infections determined significantly is more like tot be implemented.‘ In fact (and my advice): only the third sentence is needed in the text; as a rule of thumb: Statistics is a tool to help making conclusions, statistics is not the ‘result’ of a study.

All tables are not very easy to understand (but I understand the struggle to present the data). However important is that the number of responses (answers) is never given. It is not very useful (or common) to present the X2 (can you use upper case X in the text?) values, they relate to the number of responses and showing the numbers (and percentages) gives the readers much more clarity and information. Is it possible to summarize all responses in a few combined tables? -including the numbers and percentages?

The conclusion contains: ‘CAUTI prevention is multifactorial, and the only effective measure consists in the introduction of a set of preventive measures and their checks.’ Education and training (and or specific privileging) are indirectly mentioned in the remainder of the conclusions paragraph, but are they not, based on the result of the questionnaire, much more important? Also for the discussion is the questionnaire applicable (responsive) for auditing and is auditing the recommended (by the authors) tool to improve this element of health care quality. (And or does the monitoring of the incidence of CAUTI play a role? As a baseline and or a continuous care quality evaluation parameter?

Author Response

Response to Reviewer 3 Comments

Point 1:

The word facilitation in graph 1 and text and also the question are difficult to understand; is this a ‘fact’ or an ‘opinion’ and how does this relate to local and national regulations? But also: What are ‘general nurses’? are these ‘basic level’ nurses? (that would not be allowed to indicate or perform urinary catheterisation without supervision by law in many healthcare systems.) And furthermore it is not at all clear who were the responders: [cited:] ‘we distributed questionnaires to leading non-physicians (general nurses) in the positions of head, ward, and leading nurse.’ But who responded?? Were all respondents ‘identical level’? (and discuss how did (or can) the professional level of the respondents have affected the results?

Response 1: The designation of the type of hospital is based on the valid legislation of the Czech Republic; the types of hospitals are given by the Ministry of Health of the Czech Republic and the Institute of Health Information and Statistics of the Czech Republic. The hospitals, or the designation of hospital type, depend on the founder of the hospital. The designation of the profession “general nurse” is governed by the valid legislation of the Czech Republic, “general nurse” is a nurse with a university degree - 3 years (6 semesters) in a Bachelor’s degree study programme. A general nurse is competent to care for urinary catheters in patients of all ages, including performing bladder lavages, and evaluating the patient’s condition. In the Czech Republic, a general nurse is competent to perform bladder catheterisation in women and girls older than 3 years, and after obtaining special professional competence (certified courses) or specialised “intensive care” competence, he or she is competent to perform catheterisation in men - SPECIFIED IN SECTION 2.1.

In the Czech Republic, according to the legislation “basic level nurses” are called “practical nurses” - secondary school education. Practical nurses cannot perform catheterisations, so they were not respondents in the survey, and were not interviewed.

The respondents of our research were general nurses in the position of middle management - heads nurses of departments, station nurses, leading nurses, and nursing care assistants – STATED IN SECTION 2.1

Point 2:

Also was the total number of respondents 186 or 176 (internist + surgical) and or why is this 186(or 176)/194 considered low level of willingness in the discussion? Or does this refer to a low number of responses (see below: number of responses not given.)

Response 2: REVISED IN THE TEXT on page 4-5: A total of 186 respondents participated in the research, i.e., completed all three questionnaires. According to the type of ward – there were 111 (n = 111) respondents from the internist ward type, 65 (n = 65) from the surgical ward type, and 10 respondents were in the position of nursing care assistant. Thank you very much for drawing our attention to this!

TEXT FORMULATION REVISED AND SUPPLEMENTED: page 3 of the text - 194 hospitals providing acute inpatient care in the Czech Republic were contacted, 34 hospitals agreed with the survey.

Point 3:

There is a lot of repetition in the text: I give one example but almost all paragraphs are identical: ‘The same finding was made in the case of the evaluation of the existence of a standard for the prevention of urinary tract infections and the performance of regular audits of acceptance of preventive CAUTI measures. A statistically significant relationship was demonstrated between whether the workplace has implemented a standard of urinary tract infection prevention and regular audits of urinary tract infection prevention (x2= 23.171 p<0.001). Where regular audits of the prevention of urinary tract infections are performed, a standard of prevention of urinary tract infections determined significantly is more like tot be implemented.‘ In fact (and my advice): only the third sentence is needed in the text; as a rule of thumb: Statistics is a tool to help making conclusions, statistics is not the ‘result’ of a study.

Response 3: Revised based on your comments in the parts of the text 3.1. - 3.8

Point 4:

All tables are not very easy to understand (but I understand the struggle to present the data). However important is that the number of responses (answers) is never given. It is not very useful (or common) to present the X2 (can you use upper case X in the text?) values, they relate to the number of responses and showing the numbers (and percentages) gives the readers much more clarity and information. Is it possible to summarize all responses in a few combined tables? -including the numbers and percentages?

Response 4: Partially revised, some of the tables have been left in their original form for the sake of clarity and conciseness of the communication, they supplement the text appropriately.

Point 5:

The conclusion contains: ‘CAUTI prevention is multifactorial, and the only effective measure consists in the introduction of a set of preventive measures and their checks.’ Education and training (and or specific privileging) are indirectly mentioned in the remainder of the conclusions paragraph, but are they not, based on the result of the questionnaire, much more important? Also for the discussion is the questionnaire applicable (responsive) for auditing and is auditing the recommended (by the authors) tool to improve this element of health care quality. (And or does the monitoring of the incidence of CAUTI play a role? As a baseline and or a continuous care quality evaluation parameter?

Response 5: We have revised the Conclusion based on your recommendation.

Reviewer 4 Report

In this manuscript, authors conducted a survey about the practice of CAUTI prevention among hospitals in Czech Republic. Among 194 hospitals, non-physician management of 34 hospitals responded, and 186 responses were included. It seemed there was significant variation in their response, but there was no clear description of how they varied. I did not understand the importance of this study, and what this study will add to the knowledge body of evidence. There are many points which needs to be improved.

  1. Introduction – the introduction part is very long and some parts are repetitive. It can be simplified.
  2. Introduction – There should be the statement about what is not known (knowledge gap) and why authors conducted this study to address the knowledge gap.
  3. Materials and methods – In material section, first half are aim statement and should be included in the introduction part (this article ….. and We also wanted to determine….)
  4. Methods – they should outline the contents of their survey. If possible, a short description about why those questions were chosen should be included.
  5. Methods – Ethical statement of this study should be included.
  6. Results – I think it more important to show the number of responses or score the response (for example, 1 for the worst response to 5 for the best response) rather than showing the chi-square results. Chi-square results just tell us the responses were significantly different, but without further description we cannot know how they varied.
  7. Discussion – The first 3 paragraphs were summary of previous studies and did not include the content in this study. They should be removed or incorporated in introduction.
  8. Discussion – What is the clinical implication of this study? What is the next step?
  9. Authors should avoid unnecessary self-citations.

Author Response

Response to Reviewer 4 Comments

Point 1:

In this manuscript, authors conducted a survey about the practice of CAUTI prevention among hospitals in Czech Republic. Among 194 hospitals, non-physician management of 34 hospitals responded, and 186 responses were included. It seemed there was significant variation in their response, but there was no clear description of how they varied. I did not understand the importance of this study, and what this study will add to the knowledge body of evidence. There are many points which needs to be improved.

Response 1: TEXT FORMULATION REVISED AND SUPPLEMENTED: page 3 of the text - 194 hospitals providing acute inpatient care in the Czech Republic were contacted, 34 hospitals agreed with the survey. The hospital management distributed links to the questionnaires to general nurses in the positions of station nurse, head nurse, leading nurse, and nursing care assistant. A total of 186 respondents participated in the research, i.e., completed all three questionnaires.

Point 2:

Introduction – the introduction part is very long and some parts are repetitive. It can be simplified.

Introduction – There should be the statement about what is not known (knowledge gap) and why authors conducted this study to address the knowledge gap.

Response 2: The “starting point” text has been shortened, and justification for the research has been added. Information that is directly related to the research intention, and is necessary for the complexity of the article, has been left in.

Point 3:

Materials and methods – In material section, first half are aim statement and should be included in the introduction part (this article ….. and We also wanted to determine….)

Response 3: REVISED, specified according to the recommendation.

Point 4:

Methods – they should outline the contents of their survey. If possible, a short description about why those questions were chosen should be included.

Methods – Ethical statement of this study should be included.

Response 4: Revised as recommended. The ethical statement has been moved from Section 5.1 to the “Methods” section as recommended.

Point 5:

Results – I think it more important to show the number of responses or score the response (for example, 1 for the worst response to 5 for the best response) rather than showing the chi-square results. Chi-square results just tell us the responses were significantly different, but without further description we cannot know how they varied.

Response 5: Partially revised, some of the tables have been left in their original form for the sake of clarity and conciseness of the communication, they supplement the text appropriately

Point 6:

Discussion – The first 3 paragraphs were summary of previous studies and did not include the content in this study. They should be removed or incorporated in introduction.

Discussion – What is the clinical implication of this study? What is the next step?

Response 6: REVISED, specified according to the recommendation. The output of the study and a proposal for nursing practice have been added.

Point 7:

Authors should avoid unnecessary self-citations.

Response 7: One self-citation was used in the text - left in due to the research intension of the authors, the article was the starting point for launching the research in the Czech Republic.

Round 2

Reviewer 3 Report

Some of the redundancy is removed and the tables are much better, good work! 

Minor comments: specify: 'health care workers' in L22. L162: intension > intention. L 168 seems unfinished L404-405 2x 'again'

However the discussion is still very long with repetitions of the introduction and also the conclusion is more a summary. It never becomes clear (especially not in the conclusion (including in the abstract) what the authors consider the relevant result(s) of the study. (see line 615-618... which elements of implementation are good and which not and L682-684 (also vague)). In general, which elements of implementation are achieved (at an acceptable level) and which elements not, and which elements should be better implemented? It will in this regard be helpful to state the goal of the research in the introduction: 'We have done a study with questionnaires to evaluate implementation with the ultimate goal to ................' 

Author Response

Recommendations elaborated, abstract and conclusion modified. Discussion shortened and clarification added. Duplicate infoirmation removed. Text edited - unfinished sentence..

Reviewer 4 Report

In this revised manuscript, the flow seemed better, but I still had a hard time because this contains so many components without emphasis on any of the component. Did the authors want to describe the areas which needed to be improved? or did they want to show the variation of CAUTI preventive measures conducted in each hospital? I think it better to focus on several components which authors think the most important (and make discussions for those components) and keep others minimal - that will decrease the total amount of manuscript and put emphasis on most important components.

In addition, I still recommend to show the relative frequency of respondents in each category in tables 2, 4, 7 and 9. The result of chi-square test can be added to contingency table, if needed. By doing so, readers can easily see where variation exists. 

Author Response

Abstract, discussion and conclusion modified. Comments settled. Major research findings highlighted. The paper contains the results of all the research that focused on the CAUTI area. Therefore, in the correction of the text, the outputs have been modified. 
The results of the article are currently used by hospital management to modify or implement prevention programs. 
Tables 2, 5, modified as recommended. Table 9 removed - relative numbers of responses are in Table 8. x2 results reported in text. Table 7 retained for clarity of results presented. 

Translated with www.DeepL.com/Translator (free version)